# Learning with Analogical Reasoning for Robust Few-Shot Learning

## Abstract

Few-shot learning (FSL) is challenging due to limited support data for model training. The situation is much worse when the support data is contaminated with noise. To address this issue, we propose a novel **T**ransformer-based **A**nalogical **R**easoning model for **N**oisy **F**ew-**S**hot learning (TarNFS), by mimicking the human's ability of learning by analogy. Concretely, we assume the existence of a large human cultivated or AI-powered knowledge base, and hypothesize that similar concepts in the knowledge base are visually similar in the latent space as well. Then we design a transformer-based analogical reasoning model to utilize inter-concept connections among these concepts, aiming to build robust and discriminative classification boundaries. In addition, we propose a task-level contrastive learning to analogically learn from negative tasks to facilitate training with noisy tasks. Experiments demonstrate that our TarNFS enables more effective learning from limited and imperfect data. It not only improves the generalization ability of FSL in different noisy settings but also achieves competitive performance in the common clean FSL settings. Code is publicly available here.

## 1 Introduction

Modern deep learning methods have achieved great success thanks to the huge amount of data available for model training. Yet, these methods are faced with challenges in the scenario where training data is scarce. To alleviate the problem, researchers have recently resorted to few-shot learning (FSL), a task that is officially formalized as a $N$-way $K$-shot recognition problem with each of the $N$ classes having $K$ labeled images, tackling the task from various perspectives like data augmentation or hallucination, meta-learning for fast knowledge transfer, metric-learning to construct a transferable latent space, etc (Vinyals et al., 2016; Snell et al., 2017; Sendera et al., 2023; Guo et al., 2022). While advancements are made, most FSL methods are unconsciously devised to learn from clean data (as in Figure 1(a)), assuming an ideal scenario where all samples in the support set are deliberately selected to represent their class. This assumption is in sharp contrast to real-world settings where even carefully annotated and curated datasets often contain mislabeled samples (as in Figure 1(b)) due to ambiguity, automated data collection, or human error. Liang et al. (2022) have shown that existing FSL methods are quite vulnerable to such label noise.

Sample selection was proposed to handle noisy labels in FSL. By identifying and selecting potentially clean data from noisy data, these methods aim to build better prototypes using the chosen clean samples (Que & Yu, 2024). Instead of directly discarding noisy data, some leverage sample weighting to intentionally take the noisy data into consideration during learning (Killamsetty et al., 2020). Specifically, they design a weighting policy to assign higher weights to the clean data and lower weights to the noisy, in which manner the clean data are supposed to contribute more to the final prototypes and the side information in the noisy are properly utilized.

Despite their attainments, the aforementioned efforts have been substantially devoted to acquiring data-efficient strategies for constructing better category prototypes out of corrupted intra-class samples, rarely exploring the categories' connections to the open world. Differently, we human learn new concepts not only from instructions or demonstrations, but also from their rich relations with other known concepts (Brown & Kane, 1988). For example, when we encounter a zebra for the first time, it is not uncommon for us to interpret the species as "a horse with black and white striped pattern". We would also take one step further to guess that a zebra might likewise appear in zoo.

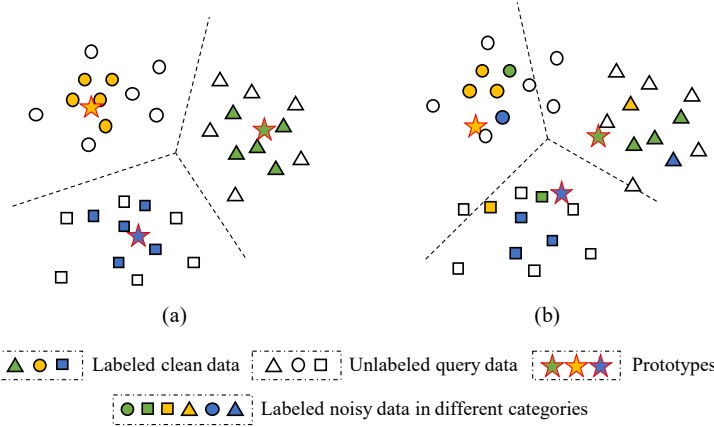

Figure 1: **FSL (left) vs FSL with noisy labels (right).** A 3-way 5-shot toy task. Noisy data in the support set would result in inaccurate prototypes and mislead a model to wrong predictions.

This ability of learning by analogy, or analogical reasoning, has been proved a fundamental building block in human learning process (Gentner & Holyoak, 1997; Thibaut et al., 2010; Bartha, 2024; Zhou et al., 2019). It ensures us to learn fast and faithfully from few examples while not being easily deceived by distracting misinformation.

To mimic the behavior, in this work we propose a novel **T**ransformer-based **A**nalogical **R**easoning model for **N**oisy **F**ew-**S**hot learning (TarNFS). Specifically, we assume a knowledge base, *e.g.*WordNet, that contains lots of connections among concepts, which represents how we human perceive the world. Given a novel concept, we find its relations with known concepts in the knowledge base to build a semantic analogy graph, and hypothesize that the relations can be analogically applied to the visual space for feature learning. Thus, unlike previous methods that struggle to learn robust category prototypes from few corrupted samples, we propose to learn with analogical reasoning to distill discriminative category boundaries, by taking the noise-independent concept connections as effective clues. The transformer architecture of TarNFS enables a natural manner of processing arbitrary number of related concepts. A task-level contrastive learning is also presented to learn from totally different few-shot tasks. This enables a task-level analogical reasoning, behind which the intuition is that two tasks should be different in the latent visual space if they have different concepts. Experiments on the MiniImageNet and TieredImageNet datasets show that, with analogical reasoning and task-level contrastive learning, our TarNFS outperforms previous SOTAs on FSL with noisy labels (*e.g.*obtain 8.3% relative improvement on MiniImageNet with 40% symmetric noise). Notably, when tested on FSL with clean data, TarNFS using Conv4-64/ResNet-12 is able to achieve 60.75%/70.29% 1-shot accuracy, performing competitively against other FSL methods.

## 2 RELATED WORK

### 2.1 FEW-SHOT LEARNING

Existing FSL approaches can be roughly categorized into four groups, *i.e.*hallucination-based, optimization-based, parameter-generating based, and metric-learning based, according to recent investigations (Wang et al., 2020; Song et al., 2023a). Hallucination-based approaches learn to estimate the distributions of novel categories (Hariharan & Girshick, 2017; Luo et al., 2021; Guo et al., 2022). Methods in this group generate new instances by sampling from the estimated distributions, turning FSL into an easily resolved many shot learning problem. Optimization-based methods (Finn et al., 2017; Nichol et al., 2018; Zhao et al., 2020) perform rapid adaption with a few training samples by learning a good optimizer or learning a well-initialized model. Parameter-generating methods follow the paradigm of hypernetworks (Ha et al., 2016; Andrychowicz et al., 2016), where the weights of the learner (often the classifier) are generated by another hypernetwork conditioned on the few samples of novel classes so that the learner can be rapidly adapted to recognize new categories (Gidaris & Komodakis, 2019; Bateni et al., 2020; Sendera et al., 2023). Metric-learning

based methods tackle FSL differently by learning to compare two examples (Sung et al., 2018; He et al., 2020a) or by learning an embedding space constrained on a selected metric (Snell et al., 2017; He et al., 2022a; Cheng et al., 2023). The main idea is to project instances into an embedding space and utilize the learned or selected metric to estimate distances from a query to candidate categories for classification. The learning often accommodates the episodic training proposed by Vinyals et al. (2016) so that the embedding space has the merit of easily generalizing to new tasks.

## 2.2 Noisy Label Learning

Noisy label learning addresses the challenge of training on datasets that contain mislabeled examples or noises. To this end, some methods focus on estimating the latent noisy transition matrix (Liu & Tao, 2016), aiming to understand and model the confusion between classes induced by label noise. In an orthogonal direction, some methods, like MentorNet (Jiang et al., 2018), prioritize learning from samples that are more likely to be correctly labeled by following certain sample weighting scheme. For instance, Co-teaching (Han et al., 2018) proposes a strategy to train two networks simultaneously, with each network providing samples it deems correctly labeled to the other for training. Li et al. (2019) leverage meta-learning to learn from synthetic noisy labels that simulate the actual training. However, these methods filter out or purify corrupted labels based on a large amount of clean data, making them unsuitable for noisy label learning in few-shot setting. To tackle the problem, Mazumder et al. (2021) propose to create hybrid features and utilize clustering approach like k-means to build more accurate prototypes. TraNFS (Liang et al., 2022) learn to automatically promote refined prototypes out of corrupted support samples by leveraging a transformer module. In this study, we leverage inter-concept connections that are noise-independent for noisy FSL. By transferring the connections among the novel and the known categories from semantic space to the visual feature space, we manage to build discriminative and noise-resistant prototypes.

## 2.3 Contrastive learning

Contrastive learning inherently learns an embedding space in a self-supervised manner where instances in positive pairs are close to each other and instances in negative pairs stay apart (He et al., 2020b; Khosla et al., 2020; Chen et al., 2020; He et al., 2022b). Its efficacy in representation learning has inspired some applications in FSL (Yang et al., 2022; Gidaris et al., 2019; Shirekar et al., 2023; Song et al., 2023b). For instance, Gidaris et al. (2019) introduced a rotation-based self-supervised pretext task into FSL and utilized the corresponding auxiliary loss for model optimization. Shirekar et al. (2023) designed an end-to-end framework in which they proposed a self-attention based message passing contrastive learning method to facilitate representation learning for unsupervised FSL. Yang et al. (2022) performed nearest centroid classification on two different views of the same FSL task and adopted the contrastive loss to overcome bias between views, which improves the transferability of representations. One observation is that most FSL methods that leverage contrastive learning actually perform instance-level contrastive constraint, while the task-level contrastive learning is overlooked. In this work, we propose to leverage task-level contrastive learning for FSL. Our method takes two views of the same task as positive and the tasks in an extra queue as negative. By aggregating each task into a task-level representation, we propose a task-level contrastive learning to push tasks with different categories away from each other and pull tasks with the same categories close to each other for model regularization during FSL.

## 3 PRELIMINARIES

**Few-Shot Learning.** Given a $N$-way $K$-shot learning task with each category having $K$ instances, FSL aims to learn to recognize the $N$ novel categories $\mathcal{C}_{\text{novel}}$ based on the few annotations. To tackle this challenging problem, we construct a meta-training set $\mathcal{D}_{\text{train}} = \{(\mathcal{D}_{\text{support}}, \mathcal{D}_{\text{query}})_i\}$ by sampling abundant fake $N$-way $K$-shot tasks out of a base dataset $\mathcal{D}_{\text{base}} = \{(\mathbf{x}_i, y_i)|y_i \in \mathcal{C}_{\text{base}}\}$ that contains large-scale training samples from $|\mathcal{C}_{\text{base}}|$ known categories (e.g., about hundreds of samples per category), and learn to learn transferable knowledge from $\mathcal{D}_{\text{train}}$ by following the meta-learning paradigm. For each task $\mathcal{T} = (\mathcal{D}_{\text{support}}, \mathcal{D}_{\text{query}}) \in \mathcal{D}_{\text{train}}$, we first randomly sample $N$ categories from the known categories $\mathcal{C}_{\text{base}}$ and $K$ instances per category as the support set $\mathcal{D}_{\text{support}} = \{(\mathbf{x}_i, y_i)\}_{i=1}^{N*K}$. Then, extra $M$ instances are randomly picked from the categories to form the query set $\mathcal{D}_{\text{query}} =$

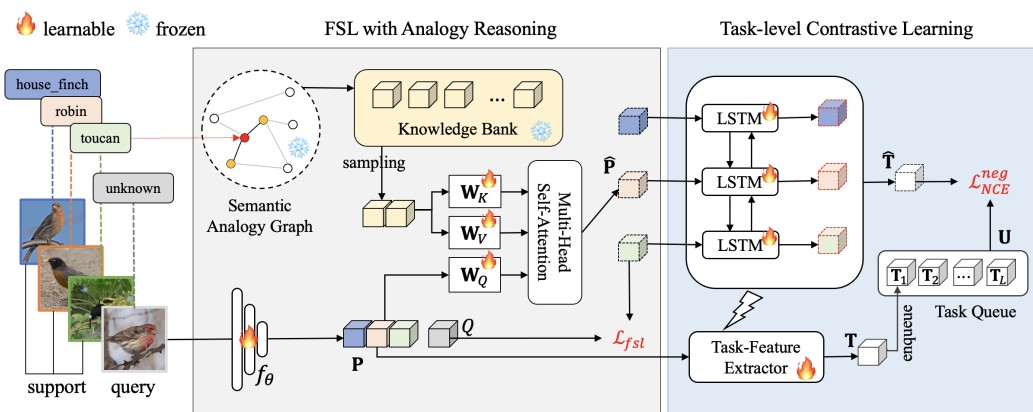

Figure 2: An overview of our Transformer-based Analogical Reasoning model for noisy FSL.

$\{(\mathbf{x}_i, y_i)\}_{i=1}^{M}$. Note that, no instance is shared between $\mathcal{D}_{\text{support}}$ and $\mathcal{D}_{\text{query}}$. A model that is capable of tackling all the tasks in $\mathcal{D}_{\text{train}}$ is considered talented when encountering a new FSL task.

**Nearest Class Mean Classifier.** Snell et al. (2017) proposed to label a query instance to the nearest category for classification. The nearest class mean (NCM) classifier is formulated as follows:

$$\hat{y} = \arg\min_c \alpha(\mathbf{f}_x, \mathbf{w}_c), \tag{1}$$

where $\mathbf{f}_x = f_\theta(\mathbf{x}) \in \mathbb{R}^d$ is the feature embedding of query instance $\mathbf{x} \in \mathcal{D}_{\text{query}}$, $\theta$ denotes learnable parameters of the learner, $\alpha$ is the distance metric, and $\mathbf{w}_c \in \mathbb{R}^d$ is the classifier weight for category $c$ which is deterministically represented as the mean of support instances belonging to the category

$$\mathbf{w}_c = \frac{1}{K} \sum_{(\mathbf{x}_i, y_i) \in \mathcal{D}_{\text{support}}} \mathbb{I}(y_i = c) f_\theta(\mathbf{x}_i). \tag{2}$$

Simple yet efficient, the NCM classifier is widely adopted in FSL. Its weights are also termed category prototypes and used interchangeably hereafter in this work.

**Few-shot Learning from Noisy Labeled Data.** NCM is vulnerable to noisy labeled data. For $\mathcal{T} = (\mathcal{D}_{\text{support}}, \mathcal{D}_{\text{query}})$, let $\kappa(\cdot)$ denote a noise sampling function that injects noise into the support set, we have its noisy counterpart $\mathcal{T}' = (\mathcal{D}'_{\text{support}}, \mathcal{D}_{\text{query}})$, where $\mathcal{D}'_{\text{support}} = \kappa(\mathcal{D}_{\text{support}})$. According to Equation (2), it is not difficult to recognise that the noise in $\mathcal{T}'$ would contaminate the weights of NCM classifier, resulting in poor or degraded performance (Liang et al., 2022). To study this problem, we follow Liang et al. (2022) to consider two types of noise sampling processes, *i.e.*the symmetric label swap noise (Van Rooyen et al., 2015) and the paired label swap noise(Han et al., 2018),to construct noisy FSL tasks. The **symmetric label swap noise** draws mislabeled samples for a category, uniformly and randomly, from the other $N-1$ categories in the task, keeping the number of total instances in each category unchanged (*i.e.*$K$). The **paired label swap noise** assumes each category has a certain matched easily confusing category. It always draws mislabeled samples for a category from its matched confusing category, simulating the real-world tendencies that one easily confuses certain classes with others during data curation. A restriction of the two noise sampling processes is that noisy samples should never tie or outnumber the clean samples in a category after noise injection, so that the NCM classifier does not collapse.

## 4 APPROACH

To facilitate FSL from noisy labeled data, as shown in Figure 2, we propose a novel method named TarNFS that leverages inter-concept connections from the novel to the known to construct robust and discriminative category prototypes. We further devise a task-level contrastive learning that pulls similar tasks close to each other and pushes dissimilar tasks far apart, to regularize model optimization from an extra task-level perspective.

### 4.1 FSL with Analogical Reasoning

We humans learn new concept fast from few demonstrations, not only due to our powerful brains but also because of our ability to make abstract connections between the new concept and the known to perform analogical reasoning. To give FSL models the similar talent, we assume there exists a knowledge base, either human-curated or AI-powered, in which connections between concepts (including both known concepts and novel ones) can be obtained. We leverage these connections for analogical reasoning for robust FSL.

Concretely, given a novel category in $\mathcal{C}_{\text{novel}}$, we draw connections between novel categories and the known $\mathcal{C}_{\text{base}}$ in the knowledge base and represent the connections as a semantic analogy graph $\mathcal{G} = \{\mathcal{A}, \mathcal{E}\}$. After that, we sample the prior experiences of known categories from a knowledge bank $\mathbf{B} = \{\mathbf{b}_c\}|_{c \in \mathcal{C}_{base}}$ to construct more robust and discriminative category prototypes for classification. Note the knowledge bank is supposed to hold prior experiences of all known categories. For each novel category, only those related known categories are selected for FSL.

To build the knowledge bank, we start by training the learner $f_\theta$ on $\mathcal{D}_{\text{base}}$ to recognize all known categories. Then, we utilize the pre-trained learner to establish the knowledge bank as follows,

$$\mathbf{b}_c = \frac{1}{N_c} \sum_{(\mathbf{x}_i, y_i) \in \mathcal{D}_{\text{base}}} \mathbb{I}(y_i = c) f_\theta(\mathbf{x}_i), \ \ \forall c \in \mathcal{C}_{base}, \tag{3}$$

where $N_c = \sum_{(\mathbf{x}_i, y_i) \in \mathcal{D}_{\text{base}}} \mathbb{I}(y_i = c)$ is the number of samples in the known category $c$. A noisy FSL task $\mathcal{T}' = (\mathcal{D}'_{\text{support}}, \mathcal{D}_{\text{query}})$ thereafter leverages the knowledge bank for robust FSL.

According to Equation (2), the task $\mathcal{T}'$ would have drifted category prototypes due to the noise in the support set. This would dramatically hurt performance as demonstrated in Section 5. To increase the robustness to such data noise, we propose a transformer-based analogical reasoning procedure to enhance prototypes. For consistency, we denote these drifted prototypes of the noisy task as $\mathbf{P} = \{\mathbf{w}_1, \mathbf{w}_2, ..., \mathbf{w}_N\} \in \mathbb{R}^{d \times N}$ and the enhanced prototypes after analogical reasoning as $\hat{\mathbf{P}} = \{\hat{\mathbf{w}}_1, \hat{\mathbf{w}}_2, ..., \hat{\mathbf{w}}_N\} \in \mathbb{R}^{d \times N}$, respectively. For each category $c$ in $\mathcal{T}'$, we have

$$\hat{\mathbf{w}}_c^T = \mathbf{w}_c^T + \text{Att}(\mathbf{W}_Q \mathbf{w}_c, \mathbf{W}_K \mathbf{B}_c, \mathbf{W}_V \mathbf{B}_c), \tag{4}$$

where $\mathbf{W}_Q \in \mathbb{R}^{d_k \times d}, \mathbf{W}_K \in \mathbb{R}^{d_k \times d}, \mathbf{W}_V \in \mathbb{R}^{d_v \times d}$ are weights of the query, key and value projection respectively, and $\mathbf{B}_c = \{\mathbf{b}_i\}|_{i \in \mathcal{A}}$ are the set of related categories sampled out of the knowledge bank on basis of the semantic analogy graph $\mathcal{G}$. The attention aggregates these known related categories into a noise-independent feature so as to refine the drifted category prototype. Mathematically, the attention is formulated as:

$$\text{Att}(\mathbf{W}_Q \mathbf{w}_c, \mathbf{W}_K \mathbf{B}_c, \mathbf{W}_V \mathbf{B}_c) = \text{softmax}(\frac{\mathbf{w}_c^T \mathbf{W}_Q^T \mathbf{W}_K \mathbf{B}_c}{\sqrt{d_k}}) \mathbf{B}_c^T \mathbf{W}_V^T. \tag{5}$$

For classification, we compare the query image in $\mathcal{T}'$ and the refined category prototypes. This ensures that the learner converges toward a point in the model sphere where the learner outputs discriminative sample features that are aligned with the enhanced prototypes. In this way, we find the drifted category prototypes can be easily and properly refined to boost performance.

### 4.2 Task-level Contrastive Learning

As the number of clean instances in each category in noisy FSL tasks exceeds that of the noisy samples, we conclude that the drifted prototypes in Section 4.1 should form discernible boundaries that are aligned with those refined ones. To this end, we further propose a task-level contrastive learning to implicitly align the two sets of prototypes to facilitate FSL. Figure 2 illustrates that our task-level contrastive learning is composed of a task-feature extractor and a task queue. Given the drifted prototypes $\mathbf{P}$ and its refined counterpart $\hat{\mathbf{P}}$, the task-feature extractor first maps the two sets of prototypes into two task representations $\mathbf{T} \in \mathbb{R}^d$ and $\hat{\mathbf{T}} \in \mathbb{R}^d$ respectively. Then we take $\mathbf{T}$ as the positive task and take the others in the task queue as negative tasks to build a task-level contrastive loss for model optimization.

Specifically, we use Bi-LSTM as the task-feature extractor to extract task representations. For each prototype $\mathbf{w}_i$ in the drifted prototypes (or $\hat{\mathbf{w}}_i$ in the enhanced prototypes), we merge the correspond-

ing forward hidden state $\vec{\mathbf{h}}_i$ and the backward hidden state $\overleftarrow{\mathbf{h}}_i$ to get its final output, *i.e.*

$$\mathbf{o}_i = [\vec{\mathbf{h}}_i; \overleftarrow{\mathbf{h}}_i], \tag{6}$$

where

$$\vec{\mathbf{h}}_i, \vec{\mathbf{c}}_i = \overrightarrow{\mathrm{LSTM}}(\mathbf{w}_i, \vec{\mathbf{h}}_{i-1}, \vec{\mathbf{c}}_{i-1}), \tag{7}$$

$$\overleftarrow{\mathbf{h}}_i, \overleftarrow{\mathbf{c}}_i = \overleftarrow{\mathrm{LSTM}}(\mathbf{w}_i, \overleftarrow{\mathbf{h}}_{i+1}, \overleftarrow{\mathbf{c}}_{i+1}). \tag{8}$$

Note prototypes in two sets are feed into the extractor one by one in the same order. The cell states $\vec{\mathbf{c}}_i, \overleftarrow{\mathbf{c}}_i$ are not used. After obtaining the output features $\mathbf{O} = \{\mathbf{o}_1, \mathbf{o}_2, ..., \mathbf{o}_N\}$ of all prototypes, we apply max pooling to aggregate these features into a task-level representation $\mathbf{T}$ (or $\hat{\mathbf{T}}$ for the ehnhanced prototypes accordingly), and use the infoNCE loss introduced in MOCO (He et al., 2020b) for contrastive learning as follows ,

$$\mathcal{L}_{\mathrm{NCE}}^{\mathrm{info}} = -\log \frac{\exp(\mathbf{T}^T \hat{\mathbf{T}}/\tau)}{\exp(\mathbf{T}^T \hat{\mathbf{T}}/\tau) + \sum_{\mathbf{T}_i \in \mathbf{U}} \exp(\mathbf{T}^T \mathbf{T}_i/\tau)}, \tag{9}$$

where $\mathbf{U} = \{\mathbf{T}_i\}|_{i \in [1,2,...,L]}$ denotes the tasks in queue and $L$ is the length of the queue.

One issue of the infoNCE loss above is that, it means to take all tasks in the queue as negative. However, the negativity is hardly gauranteed. This is because the tasks in FSL are usually randomly sampled as stated in Section 3. These tasks concequently have a high probability of being highly related with the current $\mathbf{T}$ and $\hat{\mathbf{T}}$. Given a task in the queue that shares four categories with $\mathbf{T}$, it is not reasonable to take it as negative. To mitigate this risk, in the task-level contrastive learning we select tasks from the queue that are completely distinct from $\mathbf{T}$ (*i.e.*no category overlap) for subsequent computation. These true negative tasks are utilized to construct the contrastive loss,

$$\mathcal{L}_{\mathrm{NCE}}^{\mathrm{neg}} = -\log \frac{\exp(\mathbf{T}^T \hat{\mathbf{T}}/\tau)}{\exp(\mathbf{T}^T \hat{\mathbf{T}}/\tau) + \sum_{\mathbf{T}_i \in \mathbf{U}^-} \exp(\mathbf{T}^T \mathbf{T}_i/\tau)}, \tag{10}$$

where $\mathbf{U}^-$ denotes the set of true negative tasks that share no category with $\mathbf{T}$. Our intuition is that, if two tasks have different categories, they analogically differs in the visual feature space.

### 4.3 TRAINING AND INFERENCE

The model is trained by following a multi-stage training paradigm. We first train the encoder $f_\theta$ to perform large-scale image recognition on the base dataset. Once trained, we construct prototypes for all known categories as in Equation (3) and fill the knowledge bank with these prototypes. The episodic training then is accommodated to train the learner from abundant human-crafted FSL noisy tasks in $\mathcal{D}_{\mathrm{train}}^\kappa = \{(\kappa(\mathcal{D}_{\mathrm{support}}), \mathcal{D}_{\mathrm{query}})_i\}$. For each noisy task $\mathcal{T}' \in \mathcal{D}_{\mathrm{train}}^\kappa$, we combine the FSL recogniton risk and the contrastive learning loss for model optimization as follows,

$$\mathcal{L}_{\mathrm{total}} = \mathcal{L}_{\mathrm{fsl}} + \lambda \mathcal{L}_{\mathrm{NCE}}^{\mathrm{neg}}, \tag{11}$$

where $\lambda$ is the weight to balance the two losses and

$$\mathcal{L}_{\mathrm{fsl}} = \frac{1}{|\mathcal{D}_{\mathrm{query}}|} \sum_{(\mathbf{x}, y) \in \mathcal{D}_{\mathrm{query}}} -\log \frac{\exp(-|\hat{\mathbf{w}}_y - \mathbf{f}_x|^2/\tau)}{\sum_c \exp(-|\hat{\mathbf{w}}_c - \mathbf{f}_x|^2/\tau)}. \tag{12}$$

For inference, we take the Euclidean distance as the metric to label a query instance the same category as its nearest prototype in the refined prototypes $\hat{\mathbf{P}}$ and have

$$\hat{y} = \arg\min_c |\hat{\mathbf{w}}_c - \mathbf{f}_x|^2. \tag{13}$$

## 5 EXPERIMENTS

In this section, we conducted experiments on the commonly used MiniImageNet (Vinyals et al., 2016) and TieredImageNet (Ren et al., 2018) datasets. We first introduce our experiment setups, including the datasets, evaluation metrics, implementation details, network architecture and parameter settings. We then evaluate our proposed TarNFS by comparing it to other noisy FSL methods, and experimentally justify its efficiency and the effectiveness of its key components in boosting robustness. Additionally, we showcase the efficacy our TarNFS in conventional clean FSL settings.

## 5.1 DATASETS AND EVALUATION METRICS

**Datasets.** MiniImageNet is a subset of ImageNet with 100 classes (600 images per class) and has a total of 60k images of size $84 \times 84$. The dataset is divided into 64 training classes, 16 validation classes, and 20 test classes. TieredImageNet contains 608 classes that are sampled from 34 high-level categories from ImageNet. The 34 top categories are divided into 20 for training, 6 for validation and 8 for testing, resulting 351 training classes, 97 validation classes, and 160 test classes, respectively. TieredImageNet is much larger than MiniImageNet. It contains around 779k images, about 1,300 images per class. The high-level split ensures that the test classes are semantically distinctive enough from the train classes, which provides a more challenging and realistic setting.

**Evaluation Metrics.** Conventionally, FSL considers both 5-way 1-shot and 5-way 5-shot settings. In our study, as the noise exists, we solely consider the 5-way 5-shot setting. Once trained, we randomly sampled 10,000 tasks and consider two types of noise, *i.e.*the symmetric label swap noise (Van Rooyen et al., 2015) and the paired label swap noise (Han et al., 2018), as mentioned in Section 3, to construct noisy tasks for evaluation. We conducted experiments at noise proportions of 20%, 40%, and 60% in symmetric label swap noise setting. As for the paired label swap noise, we exclusively conducted experiment at noise proportion of 40%. This is due to that paired label swap noise with a noise proportion of 20% is identical to the symmetric label swap noise with the same noise proportion that is already studied (Liang et al., 2022), and the paired label swap noise with a noise proportion more than 50% (*e.g.*60%), would cause the noisy samples outnumber the clean, leading to ambigious class, which is too challenging to be tackled. We report the mean average top-1 accuracy $\pm$ 95% confidence interval over the 10,000 noisy tasks for comparison.

## 5.2 IMPLEMENTATION DETAILS

**Architecture of the Learner.** We follow Liange *et al*. Liang et al. (2022) to choose Conv4-64, a simple yet widly adopted convolutional network that comprises 4 convolutional blocks, as the architecture of the learner. In Conv4-64, each block contains a convolutional layer with $3 \times 3$ kernels, a batch normalization layer, a relu activation layer and a $2\times2$ max pooling layer. The feature map output by the learner is of size $64 \times 5 \times 5$. For pre-training, we append a linear classifier at the end of the learner, and train the learner on the base dataset to recognize known categories until the accuracy is saturated. As mentioned in Liang et al. (2022), the naive architecture of Conv4-64 helps emphasize our method rather than the feature extractor.

**FSL with analogical reasoning.** We discard the linear classifier and use the pre-trained learner to repersent each image as a tensor of size $64 \times 5 \times 5$. For each known category, we calculate its mean representation as in Equation (3) to initialize the knowledge bank. When a novel class is encountered, we find its top 5 related known categories in WordNet via the Leacock-Chodorow similarity. Then, the mean representations of these selected known categories are retrieved from the bank for refining the drifted prototype of the novel category. The multi-head self-attention is implemented by following Vaswani (2017). The dimensions of the query, key and vlaue in the transformer-based analogy reasonling are equally set to 64.

**Task-level Contrastive Learning.** The dimension of the hidden state of our Bi-LSTM for task feature learning is set to 256. Therefore, each task is represented by a vector of dimension 512. The length of the task queue is set to 256 to ensure that there exist abundant true negative tasks for contrastive learning. In the infoNCE loss, we set the temperature $\tau$ to 0.07. We employ the Adam optimizer with an initial learning rate of 0.001 to train our model, by following the formulation in Equation (11). The total number of traning tasks is set to 20,000, and the learning rate decays by half every 2,000 tasks. We set the weight of the task-level contrastive learning loss to 1 (*i.e.*$\lambda = 1$). All our experiments are conducted on a platform equipped with an NVIDIA RTX 3090 GPU.

## 5.3 COMPARISON WITH PRIOR NOISY FSL METHODS

We compare our method with several baseline approaches in different types of noise and in different noise proportions. The results on MiniImageNet and TieredImagenet using symmetric label swap noise and paired label swap noise are reported in Table 1 and Table 2 respectively. In the tables, *Oracle* stands for ProtoNet (Snell et al., 2017) of our implementation that knows which samples in

Table 1: **FSL with symmetric label swap noise.** 5-way 5-shot classification accuracy $\pm$ 95% confidence intervals on MiniImageNet and TieredImageNet. **Bold** numbers indicate the best results in each column. Best viewed in color.

| Model \ Noise Proportion | 0% | | 20% | | 40% | | 60% | |
|---|---|---|---|---|---|---|---|---|
| Oracle | $70.09 \pm 0.16$ | $70.92 \pm 0.18$ | $68.11 \pm 0.16$ | $68.93 \pm 0.19$ | $65.49 \pm 0.17$ | $65.80 \pm 0.20$ | $61.45 \pm 0.18$ | $60.77 \pm 0.20$ |
| Matching Networks(Vinyals et al., 2016) | $62.16 \pm 0.17$ | $64.92 \pm 0.19$ | $56.21 \pm 0.18$ | $59.20 \pm 0.20$ | $46.18 \pm 0.18$ | $49.12 \pm 0.20$ | $34.66 \pm 0.18$ | $36.80 \pm 0.19$ |
| MAML(Finn et al., 2017) | $63.25 \pm 0.18$ | $63.96 \pm 0.19$ | $53.28 \pm 0.18$ | $54.62 \pm 0.19$ | $42.58 \pm 0.18$ | $43.71 \pm 0.19$ | $31.01 \pm 0.17$ | $31.74 \pm 0.17$ |
| ProtoNet(Snell et al., 2017) | $68.27 \pm 0.16$ | $71.36 \pm 0.18$ | $62.43 \pm 0.17$ | $66.15 \pm 0.19$ | $51.41 \pm 0.19$ | $55.05 \pm 0.21$ | $38.33 \pm 0.19$ | $40.61 \pm 0.21$ |
| Baseline++(Chen et al., 2019) | $67.91 \pm 0.16$ | $71.24 \pm 0.18$ | $61.87 \pm 0.17$ | $65.58 \pm 0.19$ | $51.87 \pm 0.18$ | $55.00 \pm 0.20$ | $38.36 \pm 0.19$ | $40.19 \pm 0.20$ |
| RNNP(Mazumder et al., 2021) | $68.38 \pm 0.16$ | $71.36 \pm 0.18$ | $62.43 \pm 0.17$ | $65.95 \pm 0.19$ | $51.62 \pm 0.19$ | $54.86 \pm 0.21$ | $38.45 \pm 0.19$ | $40.63 \pm 0.21$ |
| TraNFS-2(Liang et al., 2022) | $68.29 \pm 0.17$ | $70.92 \pm 0.19$ | $64.74 \pm 0.18$ | $67.33 \pm 0.21$ | $56.14 \pm 0.21$ | $58.76 \pm 0.23$ | $42.24 \pm 0.23$ | $44.17 \pm 0.25$ |
| TraNFS-3(Liang et al., 2022) | $68.53 \pm 0.17$ | $71.17 \pm 0.19$ | $65.08 \pm 0.18$ | $67.67 \pm 0.20$ | $56.65 \pm 0.21$ | $58.88 \pm 0.23$ | $42.60 \pm 0.24$ | $44.21 \pm 0.25$ |
| IDEAL(An et al., 2023) | $68.10 \pm 0.62$ | $67.93 \pm 0.72$ | $61.70 \pm 0.73$ | $61.89 \pm 0.81$ | $48.06 \pm 0.78$ | $47.86 \pm 0.85$ | - | - |
| DETA(Zhang et al., 2023) | $67.02 \pm 0.71$ | $70.06 \pm 0.76$ | $62.42 \pm 0.72$ | $64.42 \pm 0.79$ | $52.50 \pm 0.82$ | $54.80 \pm 0.89$ | $39.19 \pm 0.86$ | $40.14 \pm 0.92$ |
| **TarNFS** (Ours) | $\mathbf{72.86 \pm 0.15}$ | $\mathbf{71.86 \pm 0.19}$ | $\mathbf{68.44 \pm 0.16}$ | $\mathbf{67.86 \pm 0.20}$ | $\mathbf{61.35 \pm 0.18}$ | $\mathbf{60.53 \pm 0.21}$ | $\mathbf{52.17 \pm 0.19}$ | $\mathbf{50.52 \pm 0.21}$ |

Table 2: **FSL with paired label swap noise.** 5-way 5-shot classification accuracy $\pm$ 95% confidence intervals on MiniImageNet and TieredImageNet. **Bold** numbers indicate the best results in each column. Best viewed in color.

| Model \ Noise Proportion | 40% | |
|---|---|---|
| Oracle | $65.49 \pm 0.17$ | $65.80 \pm 0.20$ |
| Matching Networks (Vinyals et al., 2016) | $43.53 \pm 0.17$ | $46.13 \pm 0.19$ |
| MAML (Finn et al., 2017) | $40.67 \pm 0.18$ | $41.66 \pm 0.18$ |
| ProtoNet (Snell et al., 2017) | $47.77 \pm 0.19$ | $50.85 \pm 0.21$ |
| Baseline++ (Chen et al., 2019) | $47.82 \pm 0.18$ | $50.69 \pm 0.20$ |
| RNNP (Mazumder et al., 2021) | $47.88 \pm 0.19$ | $50.91 \pm 0.20$ |
| TraNFS-2 (Liang et al., 2022) | $50.63 \pm 0.22$ | $54.82 \pm 0.24$ |
| TraNFS-3 (Liang et al., 2022) | $53.96 \pm 0.23$ | $55.12 \pm 0.24$ |
| **TarNFS** (Ours) | $\mathbf{59.05 \pm 0.18}$ | $\mathbf{57.64 \pm 0.20}$ |

the support set are mislabelled and ignores them when constructing class prototypes for FSL, which represents perfect noise rejection or sample selection as described in Section 1.

Unsurprisingly, noisy samples negatively affect all methods. When the number of noisy labels in support set grows, the performance of all methods decreases dramatically on both datasets. Our TarNFS leverages inter-concept connections in the WordNet and analogically uses these relevant concepts for denoising category prototypes, yielding better performance compared to other approaches. For example, considering the 5-way 5-shot setting on MiniImageNet with 20% symmetric noise, our method provides at least a 5.1% relative improvement over prior art TraNFS. In the setting with 40% symmetric noise, our method achieves an absolute improvement of 9.94 points over ProtoNet, *which represents a significant relative drop of 70.6% in error compared to the Oracle*. Our method also surpasses TraNFS by a margin of 9.43%/4.57% on MiniImageNet/TieredImageNet with 40% paired label swap noise. In Section 5.4, we verify that the effectiveness of our method comes from the proposed analogical reasoning and task-level contrastive learning.

Comparing the performance of our method on MiniImageNet and TieredImageNet, we further find that our TarNFS consistently performs better on MiniImageNet, while the other methods achieves higher accuracies on TieredImageNet. This is probably due to that the known categories and the novel in TieredImageNet are explicitly set to originate from different high-level concepts. The high-level split makes it not easy to find highly relevant known concepts during FSL with analogical reasoning, thereby resulting in inferior performance. Even though, our method is able to outperform other approaches in different noise settings. As can be seen in Table 1, our method not only achieves good results when evaluated upon clean tasks (of 0% noise), but also is pretty robust to achieve more gains as the proportion of noise grows.

## 5.4 ABLATION STUDIES

To understand how our proposed transformer-based analogical reasoning and task-level contrastive learning help in noisy FSL, we conduct ablation experiments on MiniImageNet.

Table 3: **Ablation study of analogical reasoning and task-level constrastive learning in noisy FSL.** 5-way 5-shot classification accuracy $\pm$ 95% confidence interval on MiniImageNet with symmetric noise. "PN": ProtoNet of our implementation. "AR": FSL with analogical reasoning. "TCL": task-level contrastive learning.

| PN | AR | TCL | 0% | 20% | 40% | 60% |
|----|----|-----|------|------|------|------|
| ✓ | | | 70.16± 0.16 | 64.36± 0.17 | 54.13± 0.19 | 40.25± 0.20 |
| ✓ | ✓ | | 71.27± 0.16 | 66.78± 0.17 | 60.06± 0.18 | 51.34± 0.19 |
| ✓ | ✓ | ✓ | **72.86± 0.15** | **68.44± 0.16** | **61.35± 0.18** | **52.17± 0.19** |

Table 4: **5-way 5-shot FSL performance at various meta-training noise ratios** on MiniImageNet.

| | 0% | 20% | 40% | 0% | 20% | 40% | 60% |
|---|----|-----|-----|------|------|------|------|
| | | ✓ | | 69.10± 0.16 | 63.56 ± 0.18 | 52.85 ± 0.19 | 39.19 ± 0.21 |
| | | | ✓ | 68.67± 0.17 | 64.85 ± 0.18 | 55.76 ± 0.21 | 41.73 ± 0.23 |
| Liang et al. (2022) | | | ✓ | 67.37± 0.17 | 63.97 ± 0.19 | 55.65 ± 0.21 | 41.63 ± 0.24 |
| | | ✓ | ✓ | 68.53± 0.17 | 65.08 ± 0.18 | 56.65 ± 0.21 | 42.60 ± 0.24 |
| | ✓ | ✓ | ✓ | 68.90± 0.17 | 65.08 ± 0.18 | 56.73 ± 0.21 | 42.69 ± 0.24 |
| | | ✓ | | **73.12± 0.15** | 68.22 ± 0.16 | 60.04 ± 0.18 | 49.55 ± 0.19 |
| | | | ✓ | 72.86± 0.15 | **68.44 ± 0.16** | **61.35 ± 0.18** | 52.17 ± 0.19 |
| | | | ✓ | 70.54± 0.16 | 66.70 ± 0.17 | 60.48 ± 0.18 | **52.89 ± 0.19** |
| **TarNFS** (Ours) | ✓ | ✓ | | 72.81 ± 0.15 | 68.21 ± 0.17 | 60.69 ± 0.18 | 51.40 ± 0.19 |
| | ✓ | | ✓ | 71.84 ± 0.16 | 67.74 ± 0.17 | 60.99 ± 0.18 | 52.52 ± 0.19 |
| | | ✓ | ✓ | 71.79 ± 0.16 | 67.41 ± 0.17 | 60.43 ± 0.18 | 51.37 ± 0.19 |
| | ✓ | ✓ | ✓ | 72.13± 0.16 | 67.76± 0.16 | 60.83± 0.18 | 52.23± 0.19 |

**Effectiveness of analogical reasoning.** From Table 3, it can be seen that our analogical reasoning leverages inter-concept connections to build more strong prototypes and achieve better results, especially upon tasks with higher noise proportions. For example, considering tasks with 60% noise, Table 3 shows that the introduction of analogical reasoning only can bring an improvement of about 11 points against ProtoNet.

**Effectiveness of Task-level Contrastive Learning.** The task-level constrastive learning provides an auxilary supervision from the task perspective. It allows the model to maintain an overall characteristic of the few-shot task, no matter how the model performs per sample in each task. Table 3 shows that, by introduction of the task-level contrastive learning, we further boost the performance by an average of 1.34 points.

**Train on Tasks with Different Noise Ratios.** A foundamental hypothesis in FSL is that we should train a model to do fast learning in the environment where it would be tested (Vinyals et al., 2016). The idea inspires that it may be helpful to train the model on noisy tasks to improve its robustness to noise. To verify this, we train our model on tasks that incorporate varying symmetric noise ratios and test it on tasks with different noise proportions. The results are listed in Table 4. As can be seen, training on tasks with a single noise proportion boosts the performance on that noise level or higher during testing, which is partially consistent with the findings in Liang et al. (2022). By combining tasks with varying noise ratios for training, we observe a performance drop on clean tasks. Consistent performance boosts on tasks with 60% noise are also observed when compared to the model trained on clean tasks. However, unlike Liang et al. (2022), the performance gains in other two noise levels are not observed in our method when training with hybrid noise levels. Contrarily, we find that training on tasks with 20% noise appears to achieve the best overall performance. Thus, we report the results of our TarNFS trained on tasks with 20% noise in Table 1 for comparison.

Note we donot train our model on tasks with 60% noise in Table 4. This is because such a high noise ratio leads to ambiguous prototypes and meaningless task representations, thereby dramatically decreasing performance. Liang et al. (2022) also found that it is not helpful to train on tasks with 60% noise. When trained and evaluated on tasks with the same noise ratio, Table 4 shows that our method can consistently surpass TraNFS. For example, our TarNFS trained on clean tasks are strong enough to obtain 68.22%/60.04%/49.55% accuracy when tested on noisy tasks with 20%/40%/60% noise, boosting the performance of TraNFS in the same setting by a margin of 4.66%/7.19%/10.36% respectively. It indicates the effectiveness of our method in tackling noisy FSL.

Table 5: **Comparisons of model efficiency.** #Parameters, forward/backward time and inference time of different models.

| Model \ Time | #param | forward/task | backward/task | training/epoch | inference time (100 tasks) |
|---|---|---|---|---|---|
| ProtoNet(Snell et al., 2017) | 0.11M | 0.0096s | 0.0027s | 5.58s | 4.31s |
| DETA(Zhang et al., 2023) | 0.21M | 0.2640s | 0.1272s | - | 45.72s |
| TarNFS (wo/TCL) | 0.13M | 0.0100s | 0.0028s | 5.86s | 4.48s |
| **TarNFS** (Ours) | 0.78M | 0.0362s | 0.0213s | 7.31s | 4.55s |

Table 6: **5-way 1-shot and 5-way 5-shot classification accuracy $\pm$ 95% confidence intervals on MiniImageNet** using Conv4-64 and ResNet-12 backbones.

| Models | Conv4-64 | | ResNet-12 | |
|---|---|---|---|---|
| | 1-shot | 5-shot | 1-shot | 5-shot |
| ProtoNet(Snell et al., 2017) | $49.42 \pm 0.78$ | $68.20 \pm 0.66$ | $60.37 \pm 0.83$ | $78.02 \pm 0.57$ |
| SimpleShot(Wang et al., 2019) | $49.69 \pm 0.19$ | $66.92 \pm 0.17$ | $62.85 \pm 0.20$ | $80.02 \pm 0.14$ |
| FEAT(Ye et al., 2020) | $55.15 \pm 0.20$ | $71.61 \pm 0.16$ | $66.78 \pm 0.20$ | $82.05 \pm 0.14$ |
| META-QDA(Zhang et al., 2021) | $56.41 \pm 0.80$ | $72.64 \pm 0.62$ | $65.12 \pm 0.66$ | $80.98 \pm 0.75$ |
| PAL(Ma et al., 2021) | - | - | $69.37 \pm 0.64$ | $84.40 \pm 0.44$ |
| tSF(Lai et al., 2022) | $57.39 \pm 0.47$ | $73.34 \pm 0.37$ | $69.74 \pm 0.47$ | $83.91 \pm 0.30$ |
| STANet(Lai et al., 2023) | $57.32 \pm 0.47$ | $73.00 \pm 0.37$ | $69.84 \pm 0.47$ | $\textbf{84.88} \pm 0.30$ |
| ESPT(Rong et al., 2023) | - | - | $68.36 \pm 0.19$ | $84.11 \pm 0.12$ |
| ALFA(Baik et al., 2024) | $57.75 \pm 0.38$ | $74.10 \pm 0.43$ | $66.61 \pm 0.28$ | $81.43 \pm 0.25$ |
| MetaDif(Zhang et al., 2024) | $55.06 \pm 0.81$ | $73.18 \pm 0.64$ | $64.99 \pm 0.77$ | $81.21 \pm 0.56$ |
| **TarNFS** (Ours) | $\textbf{60.75} \pm \textbf{0.76}$ | $\textbf{74.39} \pm \textbf{0.61}$ | $\textbf{70.29} \pm \textbf{0.78}$ | $82.09 \pm 0.55$ |

**Model Efficiency.** Table 5 shows that our method do bring in additional computational cost during training, with task-level contrastive learning accounting for more than 95% of the added cost. However, it is essential to emphasize that this computational cost is not worth mentioning when compared to other computationally intensive methods like DETA. In fact, as shown in Appendix A.3, the number of new parameters introduced by our transformer-based analogical reasoning and LSTM-based task-level representation learning becomes increasingly insignificant as the number of parameters of the learner increases. Moreover, since task-level contrastive learning is utilized only for training, our method does not experience any efficiency issue and can run as fast as ProtoNet.

## 5.5 TYPICAL FEW-SHOT LEARNING PROBLEMS

As shown in Table 4, our method is not only robust to noise in the support set, but also is effective in FSL with no noise (*i.e.* at 0% noise proportion). To further justify this merit, we propose to train and evaluate our TarNFS on clean tasks and compare it with representative methods in the literature for typical FSL. We consider both 5-way 1-shot and 5-way 5-shot problems, and take Conv4-64 an ResNet-12 as the two architectures of our learner for experiments. We train our model on MiniImageNet by following the implementaion details in Section 5.2 and test it on 600 randomly hand-crafted novel tasks for validation. Results are reported in Table 6. We can observe that, with analogical reasoning by leveraging inter-concept connections and the task-level contrastive learning, our method achieves a significant improvement under the 5-way 1-shot setting with an increase of 3%/0.45% when using the Conv4-64/ResNet-12 backbone. Whereas, in the 5-way 5-shot setting, as the prototypes generated are already sufficiently accurate, our method can only achieve a modicum of improvement. This is expected. Though, according to Table 6, we surely conclude that our TarNFS contributes a new robust FSL method to the literature.

## 6 CONCLUSION

In this paper, we propose Transformer-based Analogical Reasoning model for Noisy Few-Shot learning (TarNFS), a novel model designed to address the challenges posed by mislabeled support sets samples. The TarNFS method has the following features: (1) The analogical reasoning module leverages semantic relationships to construct noise-resilient prototypes, which are effective in both noisy few-shot learning (FSL) and typical FSL scenarios. (2) The implementation of task-level contrastive learning further optimizes the model, enhancing its generalization capabilities when faced with novel tasks. The experiments on MiniImageNet and TieredImageNet demonstrate the significant success of our method, particularly under conditions of severe data noise or data scarcity.

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

## A    APPENDIX

In this appendix, we present experiments on CIFAR-FS with ResNet12 as the learner to showcase the generalization of our method to other datasets, backbones and knowledge bases.

**CIFAR-FS** is a dataset derived from CIFAR-100[1] with images of size $32 \times 32 \times 3$. It contains 100 categories with 600 instances in each class. All the classes are split into 64, 16 and 20 for training, validation and test.

**ResNet12** is a small version of ResNet that is widely explored and exploited in few-shot learning (Pan et al., 2024; Fan et al., 2023; He et al., 2022a). Like Conv4-64, ResNet12 consists of four residual blocks. Each block contains a stack of three convolutional layers with $3 \times 3$ kernels. Each convolutional layer is followed by batch normalization and a leaky ReLU non-linearity. A convolutional skip with $1 \times 1$ kernel over the convolutional stack is used in each residual block. Following each residual block, a leaky ReLU layer and a $2 \times 2$ maxpooling layer are placed at the end for non-linearity and downsampling. The number of filters used in each block is [64, 128, 256, 512] or [64, 160, 320, 640], respectively. In our experiments, we follow An et al. (2023); Pan et al. (2024) to use the later wide architecture for fair comparison.

### A.1    Noisy FSL on CIFAR-FS

We follow An et al. (2023) to resize images to $84 \times 84$ for FSL on CIFAR-FS. Therefore, each image in CIFAR-FS is represented as a tensor of size $640 \times 5 \times 5$. WordNet does not contains all categories in CIFAR-FS which hinders our direct adoption to this dataset. To work with CIFAR-FS, we use CLIP (Radford et al., 2021) to build a pseudo knowledge base that can be utilized in our method. Particularly, for each category in CIFAR-FS, we get the representation of text "A photo of [CATEGORY]". After that, we use cosine similarity to measure the relationship between each two representations. The similarities coarsely represent how we human recognize their connections in a *large but unkown knowledge base*. We then leverage the similarities to construct robust and

---

[1]https://www.cs.toronto.edu/ kriz/cifar.html

Table 7: **FSL with symmetric and paired label swap noise on CIFAR-FS (ResNet12).** 5-way 5-shot classification accuracy $\pm$ 95% confidence intervals on CIFAR-FS. **Bold** numbers indicate the best results in each column.

| Model \ Noise Proportion | 0% | 20% | 40% | 60% | 40%(paired) |
|---|---|---|---|---|---|
| Oracle | $79.99 \pm 0.66$ | $78.12 \pm 0.68$ | $75.32 \pm 0.73$ | $69.20 \pm 0.76$ | $75.32 \pm 0.73$ |
| ProtoNet(Snell et al., 2017) | $79.99 \pm 0.66$ | $74.40 \pm 0.75$ | $61.87 \pm 0.84$ | $44.43 \pm 0.85$ | $55.91 \pm 0.84$ |
| PRWN(Bertinetto et al., 2018) | $81.43 \pm 0.67$ | $75.28 \pm 0.75$ | $53.00 \pm 0.93$ | - | - |
| RNNP(Mazumder et al., 2021) | $75.56 \pm 0.77$ | $73.01 \pm 0.81$ | $59.07 \pm 1.25$ | - | - |
| IDEAL(An et al., 2023) | $\mathbf{83.86 \pm 0.61}$ | $\mathbf{80.44 \pm 0.71}$ | $62.79 \pm 0.96$ | - | - |
| **TarNFS** (Ours) | $81.17 \pm 0.64$ | $76.59 \pm 0.72$ | $\mathbf{69.72 \pm 0.76}$ | $\mathbf{57.68 \pm 0.85}$ | $\mathbf{66.02 \pm 0.81}$ |

discriminative category prototypes by integrating prior experiences of those related known ones in the knowledge bank (as decribed in Section 4.1) for noisy FSL. The experimental results in Table 7 are averaged accuracy of 600 randomly generated test episodes with 95% confidence intervals.

Note that CLIP introduces no information leakage in our method because we merely use it to obtain category connections in a pseudo knowledge base. Thereafter, only the attained connections are utilized to retrieve experiences of known categories from the knowledge bank. We DONOT use CLIP to initialize the knowledge bank.

Table 7 shows that our method generalizes well to CIFAR-FS and ResNet12. Our method can consistently outperform compared methods like ProtoNet, PRWN and RNNP[2] in both different symmetric label swap noise settings and the 40% paired label swap noise setting. IDEAL performs better than ours in clean and 20% noisy settings, except that our method surpasses it by a large margin of 6.93% in the 40% symmetric noisy setting.

## A.2 ABLATION STUDY ON CIFAR-FS

Like on MiniImageNet, we conduct ablation experiments on CIFAR-FS to further justify the effectiveness of each component in our TarNFS. Results are listed in Table 8. As can be seen, analogical reasoning consistently boosts performance in different noisy settings. Task-level contrastive learning can further boost performance by a large margin, especially in heavily noisy settings. However, task-level contrastive learning alone hardly helps in the heavily noisy settings. By disabling analogical reasoning and sampling two tasks of the same set of categories to serve as the positive pair (*i.e.*the third row in Table 8), we investigate and demonstrate that only when combined with analogical reasoning, like in our TarNFS, can the task-level contrastive learning bring significant improvements.

Table 8: **Ablation study of analogical reasoning and task-level contrastive learning in noisy FSL on CIFAR-FS (ResNet12).** 5-way 5-shot classification accuracy $\pm$ 95% confidence interval on CIFAR-FS with symmetric noise. "PN": ProtoNet of our implementation. "AR": FSL with analogical reasoning. "TCL": task-level contrastive learning.

| PN | AR | TCL | 0% | 20% | 40% | 60% |
|---|---|---|---|---|---|---|
| ✓ | | | $79.99 \pm 0.66$ | $74.40 \pm 0.75$ | $61.87 \pm 0.84$ | $44.43 \pm 0.85$ |
| ✓ | ✓ | | $80.37 \pm 0.66$ | $74.54 \pm 0.72$ | $65.81 \pm 0.77$ | $51.12 \pm 0.82$ |
| ✓ | | ✓ | $81.09 \pm 0.67$ | $74.90 \pm 0.74$ | $63.21 \pm 0.83$ | $44.30 \pm 0.88$ |
| ✓ | ✓ | ✓ | $\mathbf{81.17 \pm 0.64}$ | $\mathbf{76.59 \pm 0.72}$ | $\mathbf{69.72 \pm 0.76}$ | $\mathbf{57.68 \pm 0.85}$ |

## A.3 MORE EXPERIMENTS ON MINIIMAGENET

To complement to experiments in Section 5.3, we use ResNet12 as the learner to additionally show the effectiveness and efficiency of our proposed method. Results are listed in Table 9 and Table 10.

---

[2]Results of PRWN and RNNP are taken directly from An et al. (2023).

Table 9: **FSL with symmetric label swap noise on MiniImageNet (ResNet12).** 5-way 5-shot classification accuracy $\pm$ 95% confidence intervals on MiniImageNet.

| Model \ Noise Proportion | 0% | 20% | 40% | 60% |
|---|---|---|---|---|
| Oracle | 80.12 | 79.03 | 77.00 | 72.38 |
| PRWN(Bertinetto et al., 2018) | 73.83 | 67.91 | 49.89 | - |
| RNNP(Mazumder et al., 2021) | 65.88 | 64.78 | 50.62 | - |
| IDEAL(An et al., 2023) | 75.26 | 70.20 | 55.73 | - |
| DETA(Zhang et al., 2023) | 81.67 | 76.58 | 65.13 | 47.60 |
| **TarNFS** (Ours) | **82.09** | **76.69** | **66.49** | **51.97** |

Table 10: **Comparisons of model efficiency (ResNet12).** #Parameters, forward/backward time and inference time of different models.

| Model \ Time | #param | forward/task | backward/task | training/epoch | inference time (100 tasks) |
|---|---|---|---|---|---|
| ProtoNet(Snell et al., 2017) | 12.42M | 0.0649s | 0.0034s | 19.34s | 5.11s |
| DETA(Zhang et al., 2023) | 12.82M | 2.148s | 0.229s | - | 254.3s |
| TarNFS (wo/TCL) | 14.06M | 0.0654s | 0.0036s | 20.461s | 5.134s |
| **TarNFS** (Ours) | 15.90M | 0.0893s | 0.0223s | 23.892s | 5.178s |

The experiments are limited to symmetric label swap noise scenario as we consider the noisy situation challenging enough to adequately validate the effectiveness of our approach. For the sake of brevity, the 95% confidence intervals are not included.

