# OpenReview forum: "Learning with Analogical Reasoning for Robust Few-Shot Learning"
_ICLR.cc/2025/Conference — ICLR 2025 Conference Withdrawn Submission_

### Official Review · Reviewer_gFtz · 2024-11-01

**Soundness:** 2
**Presentation:** 3
**Contribution:** 2
**Rating:** 3
**Confidence:** 5

**Summary:**

This paper proposes a Transformer-based Analogical Reasoning model, TarNFS, designed for robust few-shot learning (FSL) in noisy environments. It leverages analogical reasoning to improve classification by building connections among categories within a semantic knowledge base, like WordNet, and introduces task-level contrastive learning to separate distinct tasks further. The proposed approach is evaluated on MiniImageNet and TieredImageNet, showing improvements over previous FSL methods, particularly under noisy conditions, and demonstrating competitive performance on standard clean data.

**Strengths:**

- The paper is well-organized, with a logical progression from problem formulation to methodology and experimental evaluation, making it accessible and easy to follow.

- The experiments cover a variety of settings, including different noise levels, ablation studies, and comparisons with several FSL baselines, providing a solid foundation for evaluating the model's effectiveness.

- TarNFS demonstrates advantages in handling noisy data, a challenging aspect of FSL, particularly through its strategy of refining noisy prototypes using analogical knowledge.

**Weaknesses:**

- While MiniImageNet and TieredImageNet are commonly used benchmarks, the model’s real-world applicability could be tested on additional datasets, e.g. Meta-dataset.
- The model relies on an existing knowledge base (WordNet), which may limit its applicability in domains lacking such resources or where these resources are incomplete.
- The Transformer-based architecture and contrastive learning process, especially with Bi-LSTM in the task-level contrastive learning, may require considerable computational resources, which are not addressed in terms of efficiency or scalability.
- The model is tailored to the structure of WordNet for analogy-based learning. Adapting this method for domains with different types of semantic structures could present challenges, and further explanation of this adaptation process is warranted.

**Questions:**

Can the proposed method outperforms the latest Noisy FSL method DETA [1]?

[1] DETA: denoised task-adaptation for few-shot learning, ICCV 2023.

---

> ### Author Response · Authors · 2024-11-24
>
> Thank you for the insightful feedbacks. We are working on more experiments on more dataset. In the following discussions, we first address part of your concerns.
>
> **W2: Limit of applicability in domains lacking such resources or where these resources are incomplete**
>
> **A2:** Our work depends on an existing knowledge base in which the semantically analogical relationships of concepts can be utilized to refine noisy prototypes. So, yes, our work has limitation as pointed out. However, we would like to emphasize that our work can be easily generalized to domains where no human curated knowledge base is ready, by leveraging an AI-powered one. For example, we can use CLIP, BGE, text2vector or other vector models that are trained supervised or unsupervised to construct a pseudo knowledge base. These models learn from abundant corpus and, to a certain extent, capture how we human perceive the world as well.
>
> We use CLIP to build such an AI-powered knowledge base for noisy FSL on CIFAR-FS dataset. Experiments below show that AI-powered knowledge bases can be  seamlessly integrated with our method. we will clarify this compatibility  in the revision.
>
> > FSL with symmetric label swap noise and paired label swap noise on CIFAR-FS dataset.
> >| Model \ Nois Proportion |  0% | 20% | 40% | 60% | 40%(paired)|
> >| ----------------------------| ----  | ----   |  ----  | -----  | ------------- |
> >|         Oracle                     | 79.99 | 78.12 | 75.32 | 69.20 | 75.32 |
> >|   ProtoNet | 79.99 | 74.40 | 61.87 | 44.43 | 55.91|
> >| TarNFS (ours) | 81.17 | 76.59 | 69.72 | 57.68 | 66.02 |
>
> **W3: The method may require considerable computational resources, which are not addressed in terms of efficiency or scalability.**
>
> **A3:** We investigate the efficiency of our method as in the table below when we do experiments on CIFAR-FS dataset. As can be seen,  our method is as efficient as ProtoNet, especially during inference. The transformer-based architecture for AR ( the 2nd row)  and LSTM architecture for task-level representations (the 3rd row) introduce negligible amount of parameters, and slightly increase the forward/backward time from a task averaged perspective.
>
> Though LSTM in TCL increase the training time by about 20% per epoch, it introduces only 1.5% more time latency during inference when compared to ProtoNet. This is because TCL is enabled during training only and disabled once the training is finished.
>
> > FSL training/testing time and efficiency on CIFAR-FS (base model is ResNet12).
> >| Model \ Time |  #parameters | forward time/task | backward time/task | training time/epoch | testing time (100 tasks) |
> >| ----------------------------| ----  | -----  | ----   |  ----  | ---- |
> >|         ProtoNet                     | 12.42M  | 0.05276s | 0.00971s | 16.031s | 4.09s |
> >|  TarNFS (wo/TCL) | 14.06M | 0.05362s | 0.01111s | 16.1057s |  4.16s |
> >| TarNFS (ours) | 15.90M |0.07902s | 0.02096s | 19.447s | 4.15s |
>
> **W4: Adapting this method for domains with different types of semantic structures could present challenges.**
>
> **A4:** As discussed in **A1**, our method is compatible  with both human-curated knowledge base (like WordNet) and AI-powered ones. For WordNet, we get relevant concepts for analogy-based reasoning via LeaCock-Chodorow similarity. For other domains with different types of semantic structures, we could find another metric (eg cosine similarity, euclidean distance, etc) for the similar purpose. The adaptation process is straightforward.
>
> ---
>
> We use WordNet as the knowledge base in our work because it is widely utilized in the literature. It does not mean that our method can only use WordNet for noisy FSL. Other human-curated or AI-powered ones can be applied. We will make a revision to clarify the compatibility with other knowledge bases of our method.
>
> **We are doing experiments on MD. We will report the results together with comparisons to DEAT as soon as possible.**
>
> Regards

---

> > ### Comment · Reviewer_gFtz · 2024-11-26
> > **Could be further improved.**
> >
> > Thanks for the authors' response, some of my concerns are addressed. Yet, I believe the paper should be further improved following reviewers' suggestions, thus I decide to keep my score.

---

> > > ### Author Response · Authors · 2024-11-27
> > >
> > > More information that we think is helpful to address your concerns.
> > >
> > > **W1: the model’s real-world applicability could be tested on additional datasets**
> > >
> > > **A1: We have additionally done experiments on CIFAR-FS as posted in discussions with the other two reviewers. Please see the corresponding part.** Experiments on the large MD dataset is in processing. We are doing our best to fulfill the reviewer's request. We need more time.
> > >
> > >
> > > **Q:  can the proposed TarNFS outperforms the latest Noisy FSL method DETA?**
> > >
> > > **A:Yes, our method compares favorably to DETA.** We use Resnet12 as backbone, train and evaluate DETA by following the F-NCC scheme as defined in the original paper [1] on miniImagenet. We use the [official implementation](https://github.com/JimZAI/DETA). In different noisy settings, DETA serves as a strong baseline.
> > >
> > > > FSL with symmetric label swap noise on miniImagenet dataset (base model is ResNet12).
> > > >| Model \ Nois Proportion |  0% | 20% | 40% | 60% |
> > > >| ----------------------------| ----  | ----   |  ----  | -----  |
> > > >|   DETA | 81.67 | 76.58 | 65.13 | 47.60 |
> > > >| TarNFS (ours) | 82.09 | 76.69 | 66.49 | 51.97 |
> > >
> > > As can be seen, DETA performs well in low noisy settings (i.e. clean and 20%) , but is heavily impacted by noisy samples under high noisy settings. This is, in our opinion, because DETA relies on in-class and out-of-class region similarities to estimate the weights of these regions. Weights of the regions are further aggregated  into the weights of images and utilized in local compactness loss and global dispersion loss for model optimization. As noise ratio increases, the weights of regions are seriously degraded, and so is the model training.  **The results on miniImagenet are consistent with those reported in Tab2 in [1].**
> > >
> > > The table above shows that our method is slightly better than DETA under noise ratios like 0%, 20% 40%, and particularly outperforms DETA by a margin of 4.37% under 60% noise ratio, demonstrating the effectiveness of our method.
> > >
> > > **We will add DETA to Tab1 for comparison and update citation accordingly.**
> > >
> > >
> > >
> > > Also note that, **Our method is more efficiency than DETA.** Since DETA samples $k$ local regions per image, estimates the weights of these regions and undergoes A-TA or F-TA for FSL, it has very high computational complexity and causes severe inference latency. Experiments on 100 test tasks (in the table below) shows that DETA is around $49\times$ slower than ours in inference.
> > >
> > > > FSL testing time and efficiency on miniImagenet (base model is ResNet12).
> > > >| Model \ Time |  #parameters | forward time/task | backward time/task | training time/epoch | testing time (100 tasks) |
> > > >| ----------------------------| ----  | -----  | ----   |  ----  | ---- |
> > > >|  ProtoNet | 12.42M | 0.06486s | 0.00336s | 19.355s |  5.109s |
> > > >|  DETA | 12.82M | 2.148s | 0.229s | - |  254.3s |
> > > >| TarNFS (ours) | 15.90M |0.08932s | 0.02233s | 23.892s | 5.178s |
> > >
> > >
> > >
> > >
> > > Hope this discussion help address your concerns. We are open for further discussion if you have other questions.
> > >
> > > [1] DETA: denoised task-adaptation for few-shot learning, ICCV 2023.

---

> > > > ### Author Response · Authors · 2024-11-27
> > > >
> > > > Further DETA experiments on miniImagenet with Conv4-64 backbone are as follows
> > > >
> > > > > FSL with symmetric label swap noise on miniImagenet dataset (base model is Conv4-64).
> > > > >| Model \ Nois Proportion |  0% | 20% | 40% | 60% |
> > > > >| ----------------------------| ----  | ----   |  ----  | -----  |
> > > > >| MAML | 63.25 | 53.28| 42.58 | 31.01 |
> > > > >| ProtoNet | 68.27 | 62.43 | 51.41 | 38.33 |
> > > > >| TraNFS-3 | 68.53 | 65.08 | 56.65 | 42.60 |
> > > > >| **DETA** | **67.02** | **62.42** | **52.50** | **39.19** |
> > > > >| TarNFS (ours) | 72.86 | 68.44 | 61.35 | 52.17 |
> > > >
> > > > The results also justify the superiority of our method to DETA.

---

> ### Author Response · Authors · 2024-11-28
>
> We have made a revision to address your concerns. Specifically,
>
> 1) To test the model’s real-world applicability on additional datasets where the knowledge base may be incomplete or missing, we conduct experiments on CIFAR-FS. We use ResNet12 as the learner so as to additionally test the model's applicability to new architecture. CIFAR-FS has categories that are not in WordNet which hinders our fast adaption to the new scenario.  To this end, we use CLIP to build connections between categories which are then utilized as concept connections in a large but unknown knowledge base in our method. Experiments in Tab7 show that our method can easily generalize to this new challenging  situation, boosting performance especially in the heavily noisy settings. Experiments on MiniImageNet with Conv4-64 (Tab1) and ResNet12 (Tab9) also demonstrate that our method generalizes well to different architectures.
>
> 2) We conduct experiments to investigate the efficiency of our method. Tab5 and Tab10 show that our method cause insignificant computational cost during training especially when the learner is large. During inference, since TCL is discarded, our method experiences no efficiency issue and runs as fast as ProtoNet in different settings.
>
> 3) We thank the reviewer for recommend DETA for comparison. DETA relies on in-class and out-of-class region similarities to estimate the weights of these regions. Weights of the regions are further aggregated into the weights of images and utilized in local compactness loss and global dispersion loss for model optimization. As can be seen, DETA can be significantly impacted by noisy samples, especially in heavily noisy settings. This is because, in our opinion, the weights of regions deteriorate seriously as the noise ratio increases, which hinders model optimization and leads to performance degradation.
>
> We are open for discussion if you have any new concern.

---

> ### Author Response · Authors · 2024-12-03
>
> As the discussion phase is coming to the end, could you please take a look at our response and see if your concerns are properly addressed. If not, we are still open for discussion.

---

### Official Review · Reviewer_PJBs · 2024-11-02

**Soundness:** 3
**Presentation:** 3
**Contribution:** 3
**Rating:** 6
**Confidence:** 5

**Summary:**

This paper proposes TarNFS, a Transformer-based analogical reasoning model to improve Few-Shot Learning (FSL) in noisy environments. It enhances class prototypes using a knowledge base with multi-head self-attention and introduces task-level contrastive learning to improve generalization under noisy conditions.

**Strengths:**

1.	The proposed analogical reasoning mechanism effectively incorporates contextual information from a knowledge base to overcome the limitations of noisy support data.

2.	The combination of multi-head self-attention and a large-scale knowledge base results in efficient enhancement of class prototypes.

3.	The inclusion of task-level contrastive learning provides additional regularization, significantly improving model generalization.

**Weaknesses:**

1.	The details of knowledge base construction and selection for analogical reasoning are insufficient, particularly regarding applicability in different tasks.

2.	While task-level contrastive learning enhances generalization, the computational complexity is high, with no discussion on training time and efficiency.

3.	Scalability with larger or different types of knowledge bases is not discussed, which may limit the generalizability of the approach.

**Questions:**

1.	During the construction of the query set using mixup, does the correlation between the sampled image and the combined augmented image affect the model's performance and stability? For instance, do cases where they belong to the same class or are completely unrelated impact the outcome?

2.	Under time constraints, does the computational cost of this construction method lead to a significant performance improvement in the model?

3.	How sensitive is the model's performance to the parameters used in the query set construction method?

---

> ### Author Response · Authors · 2024-11-23
>
> We sincerely thank the reviewer for the helpful feedbacks. We address your comments bellow:
>
> **W1: Details of knowledge base construction and selection for analogical reasoning.**
>
> **A1:** In our method, we use a knowledge base from which we select relevant known categories for analogical reasoning to defend prototypes from noisy samples. The knowledge base we intend to use (as stated in Sec1 line080) is [WordNet](https://wordnet.princeton.edu/). For the construction of WordNet, please refer to [https://wordnet.princeton.edu/](https://wordnet.princeton.edu/). Note that other knowledge bases are also applicable in our method. For instance, when lacking of such a resource or the resource is incomplete (as the case pinpointed by [Reviewer gFtz](https://openreview.net/forum?id=jPlghr8io4&noteId=DtBGgLpYe9) ), we can use CLIP or other embedding models like BGE, text2vector to construct a pseudo knowledge base in which the connections among categories can be estimated via cosine similarity metric.
>
> To be specific, we construct a pseudo knowledge base for noisy FSL on CIFAR-FS dataset using CLIP and demonstrates that our method works well in the situation. See more details in **A3**.
>
> **W2: Discussion on training time and efficiency.**
>
> **A2:** TCL utilizes the pair of positive tasks and those negative tasks in the task queue to perform task-level contrastive learning for enhancing generalization. Since representations of negative tasks are computed in previous training iterations, the additional computation cost is from the construction of representations of positive tasks, and is negligible. We compares the forward time and backward time of ProtoNet, our TarNFS without TCL and TarNFS in the following table
>
> > FSL training/testing time and efficiency on CIFAR-FS (base model is ResNet12).
> >| Model \ Time |  #parameters | forward time/task | backward time/task | training time/epoch | testing time (100 tasks) |
> >| ----------------------------| ----  | -----  | ----   |  ----  | ---- |
> >|         ProtoNet                     | 12.42M  | 0.05276s | 0.00971s | 16.031s | 4.09s |
> >|  TarNFS (wo/TCL) | 14.06M | 0.05362s | 0.01111s | 16.1057s |  4.16s |
> >| TarNFS (ours) | 15.90M |0.07902s | 0.02096s | 19.447s | 4.15s |
>
> As can be seen, LSTM in TCL slightly increases the number of model parameters and the forward/backward time.
>
> *Note that TCL is utilized in training stage only and is dropped during inference,*  introducing no efficiency concern in inference. Thus, our TarNFS is just a little bit slower than ProtoNet but greatly boosts the performance under different noisy settings.
>
>
> **W3: Scalability with larger or different types of knowledge bases.**
>
> **A3:** Our method is compatible with other knowledge bases, both the human curated (like [YOGO](https://github.com/yago-naga/yago3))  and the AI-powered (like CLIP, BGE, text2vector etc). WordNet is a good choice for miniImagenet and tieredImagenet due to that it contains all the categories in the two datasets. For experiments on datasets that such a knowledge base does not exist or is incomplete, we can use CLIP or other embedding models like BGE, text2vector to construct a pseudo knowledge base as describe in **A1**.
>
> For example, we use CLIP to build a pseudo knowledge base that is utilized by our method on CIFAR-FS dataset. Results are as follows
>
> > FSL with symmetric label swap noise and paired label swap noise on CIFAR-FS dataset.
> >| Model \ Nois Proportion |  0% | 20% | 40% | 60% | 40%(paired)|
> >| ----------------------------| ----  | ----   |  ----  | -----  | ------------- |
> >|         Oracle                     | 79.99 | 78.12 | 75.32 | 69.20 | 75.32 |
> >|   ProtoNet | 79.99 | 74.40 | 61.87 | 44.43 | 55.91|
> >| TarNFS (ours) | 81.17 | 76.59 | 69.72 | 57.68 | 66.02 |
>
> Experiments demonstrate that our method scales to work in combination with the knowledge base well.
>
>
> **We hope that we have addressed all your concerns in our response.** A revision with changes that clarifies those discussed above is coming.

---

### Official Review · Reviewer_xZn7 · 2024-11-03

**Soundness:** 3
**Presentation:** 3
**Contribution:** 2
**Rating:** 6
**Confidence:** 3

**Summary:**

This paper addresses the challenge of Few-shot Learning with contaminated support data. To tackle this issue, the authors introduce a learning paradigm inspired by analogical reasoning. Specifically, the paper proposes a new approach to model inter-concept connections, which is then leveraged to improve inference in the presence of noisy support data. Additionally, the paper introduces a task-level contrastive learning strategy to mitigate the impact of noisy labels. To validate the effectiveness of the proposed method, the authors conduct experiments on several widely-used datasets, including MiniImageNet and TieredImageNet. The results demonstrate a significant enhancement in performance compared to existing methods.


=========================================================================
The rebuttal has addressed my concerns. I choose to increase my score.

**Strengths:**

The concept of employing semantic and analogical reasoning to improve the performance of few-shot learning with noisy labels appears promising. The writing and presentation of the methods in this paper are clear and well-articulated.
This paper conducts a thorough analysis of the method using two datasets to demonstrate the contributions of each component and to compare performance across different benchmarks.
The results obtained from the two datasets indicate a significant enhancement in performance of the proposed method compared to existing approaches.

**Weaknesses:**

While the author presents a detailed analysis using two datasets, I am concerned about their similarity. It would be beneficial if additional results could be demonstrated using other datasets, such as CUB and CIFAR-FS, to further validate the robustness of the method.
The methodology would be more convincing if it were also validated on a larger dataset, such as the Meta-Dataset. This would help in assessing its scalability and effectiveness across a broader range of learning scenarios.
The training procedure is multi-staged and could pose challenges in terms of reproducibility. Unfortunately, the author does not provide the code, which would be extremely helpful for those attempting to replicate the study's results.

**Questions:**

From Table 4, I noted that the proposed method outperforms the oracle under a 20% noise setting. Could the author provide examples of several pairs of support and query instances where the oracle fails to classify correctly, but the proposed TarNFS method succeeds? This would offer deeper insights into the strengths of the method.
I observe that the TCL module exhibits a larger margin of improvement compared to AR. Could the author present an analysis that isolates the impact of using the TCL module alone? This comparison would help clarify the individual contribution of the TCL module to the overall performance.

---

> ### Author Response · Authors · 2024-11-24
>
> We sincerely thank the reviewer for recognizing the contribution of our work. We address your concerns below.
>
> **W1: " additional results could be demonstrated using other datasets".**
>
> **A1:** As suggested, we conduct experiments on the CIFAR-FS dataset to demonstrate the effectiveness of our TarNFS. On CIFAR-FS, we note that not all categories are included in the knowledge base (i.e. WordNet) as pointed out by [Reviewer gFtz](//openreview.net/forum?id=jPlghr8io4¬eId=DtBGgLpYe9). To perform analogy reasoning, we use CLIP to estimate the similarities or connections between novel categories and known categories. Specifically, for each category, CLIP takes "A picture of [category]" as input and represents the [category] as a vector. We then use cosine similarity to find associated known categories for each novel category, which builds a pseudo knowledge base that can be alternatively utilized in TarNFS.
>
> Results are as follows:
>
> > FSL with symmetric label swap noise and paired label swap noise on CIFAR-FS dataset.
> >| Model \ Nois Proportion |  0% | 20% | 40% | 60% | 40%(paired)|
> >| ----------------------------| ----  | ----   |  ----  | -----  | ------------- |
> >|         Oracle                     | 79.99 | 78.12 | 75.32 | 69.20 | 75.32 |
> >|   ProtoNet | 79.99 | 74.40 | 61.87 | 44.43 | 55.91|
> >| TarNFS (ours) | 81.17 | 76.59 | 69.72 | 57.68 | 66.02 |
>
> Ablation study on CIFAR-FS in noisy setting below demonstrate the effectiveness of each component in our TarNFS.
>
> >| PN | AR | TCL |  0% | 20% | 40% | 60% |
> >| ----| ----| ----- | ----  | ----   |  ----  | -----  |
> >| $\surd$| | | 79.99 | 74.40 | 61.87 | 44.43 |
> >| $\surd$ | $\surd$|  | 80.37| 74.54 | 65.81 | 51.12 |
> >| $\surd$ | $\surd$| $\surd$ | 81.17 | 76.59 | 69.72 | 57.68 |
>
> We donot have enough time to complete experiments on Meta-Dataset yet. MD is of size 200GB. Moreover, the noisy setting on MD is different from those previously studied in the literature[1-2].
>
> Since our experiments on miniImagenet, tieredImagenet and cifar-fs datasets showed the effectiveness of our method, we hypothesize that our method works on MD too.
>
> [1] Liang, Kevin J., et al. "Few-shot learning with noisy labels." CVPR. 2022.
>
> [2] Li, Junnan, et al. "Learning to learn from noisy labeled data." CVPR. 2019.
>
> **W2: Code for reproducibility**
>
> **A2:** To facilitate reproducibility, we will make the code public available on github. For double-blind review, our code can be temporarily found [here](//anonymous.4open.science/r/iclr2088).
>
> **Q: TarNFS outperforms the oracle under a 20% noise setting.**
>
> **A:** Oracle stands for ProtoNet of our implementation that only uses clean support images for FSL ( as described in Sec5.3). As listed in Table1 in Sec5, our TarNFS not only outperforms *Oracle* under clean FSL setting, but is also slightly better than *Oracle* under 20% noisy FSL setting on miniImagenet. This is due to that our TarNFS can learn more robust prototypes be leveraging analogy reasoning. But we do not insist that our method can consistently surpass *Oracle* in noisy settings. We note that the performance of FSL is indeed seriously impacted by noisy samples in the support set. In noisy settings, our TarNFS is generally inferior to *Oracle* as expected.
>
> As requested by the reviewer, we here provide a task with 20% noise ratio that our method performs better than *Oracle*. [Click to view](//anonymous.4open.science/r/iclr2088/assets/task_sample.png).
>
> Note: we do not indicate any insights by the illustration above, nor do we encourage readers or reviewers to hallucinate so. This is because FSL must be tested on a bunch of tasks. The performance on one task means nothing.
>
> **Q: Impact of using the TCL module alone.**
>
> **A:** In our TarNFS, we use prototypes before and after AR to construct a pair of positive task representations for TCL. *It is difficult to analyze the impact TCL without considering AR*. To isolates the impact of TCL as requested, we sample two FSL tasks from the same set of categories as a pair of positive tasks and disable the AR component. Results on CIFAR-FS w/wo AR are as follows:
>
> > FSL with symmetric label swap noise on CIFAR-FS dataset.
> >| Model \ Nois Proportion |  0% | 20% | 40% | 60% |
> >| ----------------------------| ----  | ----   |  ----  | -----  |
> >|         Oracle                     | 79.99 | 78.12 | 75.32 | 69.20 |
> >|   ProtoNet | 79.99 | 74.40 | 61.87 | 44.43 |
> >|  **TarNFS(wo/AR)**| **81.09** | **74.90** | **63.21** | **44.30** |
> >| TarNFS (ours) | 81.17 | 76.59 | 69.72 | 57.68 |
>
> As can be seen, TCL alone can boost performance when compared to ProtoNet, especially under clean setting and 40% noisy setting. And, our TarNFS that combines AR and TCL obtains much more better results in all settings.
>
>
> ---
> Hope we have addressed your concerns. **More experiments on MD is in processing. We will give a revision as soon as possible.**

---

> > ### Author Response · Authors · 2024-11-28
> >
> > We have submitted a revision in which we believe we address your concerns. Specifically,
> >
> > 1) we conduct experiments on CIFAR-FS and MiniImageNet with ResNet12 as the learner. Experiment settings (including how we extend to domains where the knowledge base is incomplete or missing) and results are in Appendix A. Experiments with new learner on new dataset prove that our proposed method is generalizable.
> >
> > 2) We investigate the efficiency of our method and compare it with other methods. The analysis of efficiency when Conv4-64 serves as the learner is presented in Tab5 in Sec5. Analysis when ResNet12 server as the learner is presented in Tab10 in Appendix. Results demonstrate that our method introduces insignificant computational cost, especially when the number of parameters in the learner is large. Our method experiences no efficiency issue and runs as fast as ProtoNet in different settings.
> >
> > 3) We additionally studied the impact of TCL from both the efficiency perspective and the effectiveness perspective. Tab5 shows that TCL accounts for more than 95\% of the newly introduced parameters. However, the additional computational overhead is negligible, both in the small-scale learner (i.e. Conv4-64) scenario where processing is inherently swift, and in the large-scale learner (i.e. ResNet12) context. Ablation studies conducted on MiniImageNet (Table 3) and CIFAR-FS (Table 8) demonstrate that TCL is also beneficial on its own, and when combined with AR, it yields even more substantial performance improvements.
> >
> > 4) We release the code for reproducibility.
> >
> >
> > Hope we address your concerns in our revision. We are open for discussion if you have any new concern.

---

### Author Response · Authors · 2024-12-03
**General Response**

To validate the effectiveness of our method on more datasets, we conduct extra experiments on CIFAR-FS and CUB-200. We also tried different learners (like Conv4-64, ResNet12, ResNet18) to show the versatility of our method. Experiments results are as follows,

----

### **Results on MiniImageNet**

On MiniImageNet, we use the human curated knowledge base WordNet to discover concept connections for robust few-shot learning.

> FSL with symmetric label swap noise on miniImagenet dataset (base model is Conv4-64).
>| Model \ Nois Proportion |  0% | 20% | 40% | 60% |
>| ----------------------------| ----  | ----   |  ----  | -----  |
>| MAML[1] | 63.25 | 53.28| 42.58 | 31.01 |
>| ProtoNet[2] | 68.27 | 62.43 | 51.41 | 38.33 |
>| TraNFS-3[3] | 68.53 | 65.08 | 56.65 | 42.60 |
>| **DETA**[4] | **67.02** | **62.42** | **52.50** | **39.19** |
>| TarNFS (ours) | 72.86 | 68.44 | 61.35 | 52.17 |

>Ablation study on MiniImageNet (base model is Conv4-64)
>| PN | AR | TCL |  0% | 20% | 40% | 60% |
>| ----| ----| ----- | ----  | ----   |  ----  | -----  |
>| $\surd$| | | 70.16 | 64.36 | 54.13 | 40.25 |
>| $\surd$ | $\surd$|  | 71.27| 66.78 | 60.06 | 51.34 |
>| $\surd$ | $\surd$| $\surd$ | 72.86 | 68.44 | 61.35 | 52.17 |

> FSL with symmetric label swap noise on miniImagenet dataset (base model is ResNet12).
>| Model \ Nois Proportion |  0% | 20% | 40% | 60% |
>| ----------------------------| ----  | ----   |  ----  | -----  |
>| PRWN[6] | 73.83 | 67.91| 47.32 | - |
>| RNNP[7] | 65.88 | 64.78| 50.62 | - |
>| PapNet[8] | 72.86 | 67.09 | 50.21 | - |
>| IDEAL[5] | 75.90 | 71.36 | 57.35 | -
>|   DETA[4] | 81.67 | 76.58 | 65.13 | 47.60 |
>| TarNFS (ours) | 82.09 | 76.69 | 66.49 | 51.97 |

----

### **Results on CIFAR-FS**

CIFAR-FS contains concepts that are not in WordNet. To cope with this, we utilize CLIP to build an AI-powered knowledge base in which the connections between two concepts are taken as the cosine similarities between them.

> FSL with symmetric label swap noise and paired label swap noise on CIFAR-FS dataset (base model is ResNet12).
>| Model \ Nois Proportion |  0% | 20% | 40% | 60% | 40%(paired)|
>| ----------------------------| ----  | ----   |  ----  | -----  | ------------- |
>|         Oracle                     | 79.99 | 78.12 | 75.32 | 69.20 | 75.32 |
>|   ProtoNet | 79.99 | 74.40 | 61.87 | 44.43 | 55.91|
>| TarNFS (ours) | 81.17 | 76.59 | 69.72 | 57.68 | 66.02 |

>Ablation study on CIFAR-FS (base model is ResNet12)
>| PN | AR | TCL |  0% | 20% | 40% | 60% |
>| ----| ----| ----- | ----  | ----   |  ----  | -----  |
>| $\surd$| | | 79.99 | 74.40 | 61.87 | 44.43 |
>| $\surd$ | $\surd$|  | 80.37| 74.54 | 65.81 | 51.12 |
>| $\surd$ | $\surd$| $\surd$ | 81.17 | 76.59 | 69.72 | 57.68 |

----

### **Results on CUB200**

As there is no existing knowledge base corresponding to CUB200, we build an AI-powered one like that on CIFAR-FS. We use ResNet18 as the learner on this dataset.

> FSL with symmetric label swap noise and paired label swap noise on CUB200 dataset (base model is ResNet18).
>| Model \ Nois Proportion |  0% | 20% | 40% | 60% | 40%(paired)|
>| ----------------------------| ----  | ----   |  ----  | -----  | ------------- |
>|         URL[9]                     | 84.11 | 80.79 | 71.01 | 52.82 | 64.80 |
>|   DETA[4] | 84.77 | 81.12 | 71.61 | 54.11 | 65.09|
>| TarNFS (ours) | 93.53 | 91.89 | 86.68 | 74.80 | 81.53 |

----

In all the experiments above, we omit 95\% confidence intervals for brevity.

We believe that these experiments can thoroughly validate the effectiveness of our proposed method and demonstrate the generalization of our work to other datasets and learners.

We made a revision to include part of the experiments so that the revision can address the reviewers' concerns properly.

----

[1] Finn et al. "Model-agnostic meta-learning for fast adaptation of deep networks." ICML, 2017.

[2] Snell et al. "Prototypical networks for few-shot learning." NeurIPS, 2017.

[3] Liang et al. "Few-shot learning with noisy labels." CVPR, 2022.

[4] Zhang et al. "Deta: Denoised task adaptation for few-shot learning." ICCV, 2023.

[5] An et al. "From instance to metric calibration: A unified framework for open-world few-shot learning." TPAMI, 2023.

[6] Bertinetto et al. "Meta-learning with differentiable closed-form solvers." arXiv preprint arXiv:1805.08136 (2018).

[7] Mazumder et al. "Rnnp: A robust few-shot learning approach." WACV. 2021.

[8] Lu et al. "Robust few-shot learning for user-provided data." TNNLS, 2020.

[9] Li et al. "Universal representation learning from multiple domains for few-shot classification." ICCV, 2021.

---

### Note · Authors · 2024-12-31

I have read and agree with the venue's withdrawal policy on behalf of myself and my co-authors.